# Zn-Catalyzed Regioselective and Chemoselective Reduction of Aldehydes, Ketones and Imines

**DOI:** 10.3390/ijms232012679

**Published:** 2022-10-21

**Authors:** Miaomiao Zhang, Hongmei Jiao, Haojie Ma, Ran Li, Bo Han, Yuqi Zhang, Jijiang Wang

**Affiliations:** Laboratory of New Energy & New Functional Materials, Shaanxi Key Laboratory of Chemical Reaction Engineering, College of Chemistry and Chemical Engineering, Yan’an University, Yan’an 716000, China

**Keywords:** zinc, reduction, aldehydes, ketones, imines

## Abstract

An operationally convenient Zn-catalyzed synthesis of alcohols by the reduction of aldehydes, ketones, and *α*,*β*-unsaturated aldehydes/ketones is reported. It is a rare example of using mild and sustainable HBpin as a reductant for catalytic reduction of carbonyl compounds in the absence of acid or base as hydrolysis reagent. The reaction is upscalable and proceeds in high selectivity without the formation of boronate ester by-products, and tolerates sensitive functionalities, such as iodo, bromo, chloro, fluoro, nitro, trifluoromethyl, aminomethyl, alkynyl, and amide. The Zn(OAc)_2_/HBpin combination has been also proved to be chemoselective for the C=N reduction of imine analogs.

## 1. Introduction

Selective reduction is one of the most efficient reactions in modern synthetic chemistry because of it is importance in the manufacture of chemicals and basic raw materials [1,2,3,4]. The reduction of carbonyl compounds is a straightforward method to prepare alcohols, which are widely used as chemical intermediates for the synthesis of natural products, pharmaceuticals, agrochemicals, and functional materials [5,6].Traditionally, the reduction of carbonyl compounds was usually accomplished by using stoichiometric amounts of metal hydrides (NaBH_4_) and Lewis acid [7,8]. Alternatively, transition-metal-catalyzed hydrogenation of C=O bond in the presence of hydrogen (as the reductant) could also provide corresponding alcohol [9,10,11,12]. In the past two decades, pinacolborane (HBpin=4,4,5,5-tetramethyl-1,3,2-dioxaborolane), as a mild and sustainable carbonyl reductant, has attracted considerable interest [13,14,15,16,17,18,19,20,21,22,23]. Metal-catalyzed hydroboration of polarized C=O bonds for the preparation of alcohols has been developed [24,25,26]; however, certain special reagents are required for the hydrolysis reaction, such as HCl [27,28], NaOH [29,30], H_2_O_2_ [31], SiO_2_ [32,33,34,35,36] (Figure 1a). In 2019, the commercially available MgBu_2_-catalyzed reduction of *α*,*β*-unsaturated ketones for the synthesis of alcohols was reported; however, the sensitive functional groups could not survive under the reported conditions [37].

To develop cost-effective hydrogenation protocols, the replacement of precious transition metals with earth-abundant, low-cost, first-row metals is contemporary and highly desirable in terms of sustainability [38,39,40,41,42,43]. Zinc, as a typical first-row transition metal, is abundant, inexpensive, and nontoxic, having been applied in a wide range of chemical reactions [44,45,46,47]. In 2006, an efficient ZnCl_2_-catalyzed selective alkylation of ketones for the synthesis of tertiary alcohol in the presence of RMgCl (1.3 equiv.) was reported [48]. Zn-catalyzed hydrogenation of carbonyl in the presence of NH_4_Cl was also reported with a wide range of substrates bearing various sensitive groups, but excessive amounts of metallic zinc were necessary [49]. Recently, Panda’s group reported that zinc(II) complexes catalyzed chemoselective hydroboration of aldehydes and ketones to achieve corresponding primary and secondary alcohols, requiring the necessary SiO_2_/MeOH as hydrolysis reagent [50]. Therefore, it is important to develop a simple and efficient zinc catalytic system for the reduction of carbonyl groups with good tolerance of functional groups.

In this work, we present Zn(OAc)_2_-catalyzed chemoselective reduction of aldehydes, ketones, and *α*,*β*-unsaturated ketones for the synthesis of corresponding alcohols dispensed with special hydrolysis steps. In addition, Zn(OAc)_2_/HBpin was not only active for C=O reduction, but also chemoselective for the C=N reduction of imine under optimized conditions (Figure 1b).

## 2. Results

We began our catalytic studies by using HBpin as a reducing agent in THF and 2-naphthaldehyde **1a** as a model substrate (Table 1). Different zinc salts were tested at 45 °C for 24 h (Table 1, entries 1–3). Treatment of 2-naphthaldehyde **1a** in the presence of HBpin (1.5 equiv.) and Zn(OTf)_2_ (10 mol%) in THF at 45 °C for 24 h gave the hydrogenated product naphthalen-2-ylmethanol **2a** in 44% yield. We found that directly hydrogenated products were obtained in the absence of acid or base as hydrolysis reagent, and the undesired hydroboration products of carbonyl groups were not observed. The hydrogenated product **2a** was isolated in 80% and 94% with ZnCl_2_ and Zn(OAc)_2_, respectively (entries 2–3). Under the same conditions, the reduction products were not detected when silane was used as a reducing agent (entries 4). It is surprising to find out that that there was no influence on the catalytic activity with significantly lowered temperature (rt) and shortened time (1 h), which is a huge advantage of the Zn-HBpin system (entries 5). Further optimization revealed that the loading of Zn(OAc)_2_ could be cut to 1 mol % while maintaining the yield of product **2a** at 93% (entries 6). Various solvents, including EtOH, toluene, DCM, and CH_3_CN, were found to be unsatisfactory (entries 7–10). The hydrogenation of **1a** was also successful with other transition metal catalysts, such as FeCl_2_, CoBr_2_, NiBr_2_, and MnBr(CO)_5_; however, the yield of **2a** was much lower (41–80%) under standard conditions (entries 11–14). Moreover, CrCl_2_ and CuCl_2_ failed to give any hydrogenation product (entries 15 and 16). It is worth noting that the Zn-catalyzed hydrogenation of **1a** proceeded smoothly in the presence of a huge excess of Hg (0) (100 equiv., entry 6, footnote d), indicating that it is highly likely to be a homogeneous catalytic system.

With the optimized reaction conditions in hand, the scope of the zinc-catalyzed reduction of aldehydes was examined. As shown in Figure 2, all substrates were well tolerated under optimal conditions, and the desired hydrogenation products were isolated in good to excellent yields. Since the hydrogenation of *o*-nitro (**2b**), *o*-methoxy (**2c**), and *m*-nitro (**2d**) aldehydes gave the corresponding primary alcohols in good yields and selectivity, it can be concluded that the position (*ortho*, *meta*, *para*) and substituent effect (electron donating vs. electron withdrawing) have no obvious influence on the efficiency of hydrogenation. A series of parasubstituted aromatic aldehydes were investigated. The results showed that various electron-donating and electron-withdrawing groups (e.g., halogen, nitro, trifluoromethyl, and aminomethyl) were all well tolerated (**2e**–**2m**). Remarkably, the substrates bearing an alkynyl and amido group were well tolerated as well. The corresponding reduction products were obtained in 65% and 93%, respectively (**2n**–**2o**). Interestingly, the subjection of 4-acetyl benzaldehyde under the optimal condition gave a reduction of a formyl group. However, increasing the loading of the hydrogen source could easily boost the efficiency to 87% with the reduction of both carbonyl groups (**2p**). The treatment of *p*-dibenzaldehyde with 1.5 equiv. of HBpin for 12 h gave product **2q** with the selective reduction of one carbonyl group. In addition, various heavily functionalized aromatic, ferrocene complex and polycyclic aromatic hydrocarbons bearing an aldehyde group were explored as well. All the reduced primary alcohol products were obtained in high yields (77–89%, **2r**–**2v**). Inspired by the promising results, the feasibility of Zn(OAc)_2_/HBpin for catalytic reduction of heteroaromatic aldehydes was further examined. As expected, various thiophene, furan, pyridine, quinoline aldehyde analogues can be regioselectively reduced and gave the corresponding primary alcohols in good yields (65–92%, **2w**–**2ab**).

Next, a series of aromatic and aliphatic ketones were examined (Figure 3). It is important to mention that Zn(OAc)_2_-catalyzed reduction of ketones would only proceed at an elevated temperature (60 °C). As expected, the boronate ester was not observed during the reaction (analyzed by ^1^H NMR).

Both the diaryl ketone **4c** and aryl alkyl ketones **4a**, **4b**, and **4d**–**4l** were efficiently converted to the corresponding secondary alcohols under optimal conditions. Various electron-donating and electron-withdrawing groups, such as bromide, chloride, fluoride, trifluoromethyl, cyano, nitro, and amino, were well tolerated. Furthermore, the efficiency of the demonstrated catalytic system was not compromised in the presence of heteroatom (**4l**). Last but not least, dialkyl ketones were also examined, and the corresponding secondary alcohols were obtained in moderate yields (79–90%, **4m**–**4o**).

Inspired by these results, the feasibility of Zn(OAc)_2_/HBpin-catalyzed reduction of *α*,*β*-unsaturated aldehyde/ketones was examined (Figure 4). The carbonyl group of *α*,*β*-unsaturated aldehydes/ketones can be regioselectively reduced at an elevated temperature (45 °C) by the formation of allylic alcohols in good yields. We found that the C=C bond reduction was not observed according to the ^1^H NMR analysis. The efficiency and selectivity were not significantly affected by the type and location of substituents in the aromatic ring, and all the corresponding allylic alcohols were obtained in fair to good yields (81–95%, **6a**–**6m**). Moreover, fluoride, chloride, bromide, iodine, trifluoromethyl, and heteroatom, which are usually sensitive to catalytic hydrogenation, were intact (**6g**–**6m**). Lastly, the subjection of alkynone to the optimal conditions gave the corresponding product **6n** in 74% with the preservation of the alkyne group.

In order to check the generality of the demonstrated Zn(OAc)_2_/HBpin system, the subjection of the gram scale of **1a** (1.56 g, 10 mmol) to the optimal conditions with an extended reaction time (8 h) gave **2a** in 94% (Figure 5).

Since it has been demonstrated that more than two carbonyl groups could be reduced with increased loading of a Zn catalyst (Figure 2, **2p**), it is important to find out the reactivity of different carbonyls in aldehydes, ketones, esters, and amides. Hence, the chemoselectivity of the Zn-catalyzed hydrogenation reaction was further explored (Figure 6). The treatment of an equal molar of aldehyde (1 mmol) and ketone (1 mmol) with Zn(OAc)_2_ (1 mol%) and HBpin (1.5 equiv.) in THF at ambient temperature for 2 h gave naphthalen-2-ylmethanol as a major reduction product (92%) with the recovery of naphthalen-2-yl methyl ketone (Figure 6a). A similar chemoselectivity was also observed for the pair of 4-methylbenzaldehyde and 4-methylacetophenone. Most of the 4-methylbenzaldehyde was reduced, while the 4-methylacetophenone was successfully recovered (Figure 6b). The treatment of acetophenone along with ethyl benzoate and *N*,*N*′-dimethyl benzamide respectively with Zn(OAc)_2_/HBpin gave the reduction product of acetophenone as the major one (90% and 91%) with the recovery of ethyl benzoate (97%) and *N*,*N*′-dimethyl benzamide (96%, Figure 6c,d). In summary, the reactivity of Zn-catalyzed hydrogenation could be concluded in the following order: aldehyde > ketone > ester ≈ amide.

Out of curiosity, the reactivity of imines was explored at the optimal conditions. It is surprising to find out that the hydrogenation products **8** were obtained in excellent yields (Figure 7). Additionally, the electron-withdrawing groups did not affect reduction efficiency at all. Thus, it is an excellent alternative method for the preparation of secondary amine.

To have an insight of the reaction mechanism, the kinetic behavior associated with Zn-catalyzed hydrogenation of 2-naphthaldehyde was explored. The hydrogenation proceeded rapidly at the first 0.5 h to give **2a** in 88% yield (Figure 1a). The yield was increased to 94% in the next half an hour. There was a significant improvement in the next hour, as the reaction was almost finished in the first hour. The dependence of the initial reaction rate on the substrate, catalyst, and HBpin was investigated. The initial reaction rate (Δ[**2a**]/Δt) for the initial concentration of **1a**, Zn(OAc)_2_ and HBpin catalysts showed a clear positive correlation, suggesting that the catalytic concentration of 2-naphthaldehyde, Zn(OAc)_2_, and HBpin determined the initial rate (Figure 1b–d).

The reduction of 2-naphthaldehyde (**1a**) with 1.5 equiv. of DBpin showed that deuterium was introduced mainly at the C2 positions of naphthalen-2-ylmethanol (92%, Figure 2a). On the basis of the experimental results and literature report [30,37,51], the proposed Zn-catalyzed reduction of carbonyl compounds is shown in Figure 2b. First of all, commercially available Zn(OAc)_2_ reacts with HBpin to in situ generate active ZnH(OAc) species [52,53,54], and the (OAc)Bpin intermediate was observed by ^11^B NMR, followed by the addition of C=O bonds of substrate **1** or **3** or **5** to provide the corresponding Zn alkoxide species. Finally, HBpin reproduces active ZnH(OAc) species by the formation of the corresponding boronic esters and provides corresponding alcohols.

## 3. Materials and Methods

### 3.1. Chemicals and Reagents

Unless otherwise noted, materials and solvents were purchased from Tokyo Chemical Industry, Aldrich Inc., Alfa Aesar, Adamas-beta, and other commercial suppliers and used as received. Zn(OAc)_2_ (99.5%) and ZnCl_2_ (98%) were purchased from Innochem and used as received. CuCl_2_ (98%) and Zn(OTf)_2_ (99%) were purchased from Adamas-beta and used as received. CoBr_2_ (97%), Mn(CO)_5_Bri (98%), CrCl_3_ (99%), and NiBr_2_ (99%) were purchased from Aldrich Inc. and used as received. FeCl_2_ (99.5%) was purchased from Aladdin and used as received. Aldehyde and ketone derivatives were purchased or were prepared according the known procedures.

All reactions dealing with air- or moisture-sensitive compounds were carried out in a flame-dried, sealed Schlenk reaction tube under an atmosphere of nitrogen. Analytical thin-layer chromatography was performed on glass plates coated with 0.25 mm 230–400 mesh silica gel containing a fluorescent indicator (Merck). Flash silica gel column chromatography was performed on silica gel 60 N (spherical and neutral, 140–325 mesh), as described by Still. NMR spectra were measured on a Bruker AV-400 spectrometer and reported in parts per million. ^1^H NMR spectra were recorded at 400 MHz in CDCl_3_, or DMSO-*d*_6_ was referenced internally to tetramethylsilane as a standard, and ^13^C NMR spectra were recorded at 100 MHz and referenced to the solvent resonance. Appendix A.

### 3.2. General Procedure for Zn-Catalyzed Reduction of Aldehydes

A mixture of corresponding aldehyde derivatives (0.2 mmol) and Zn(OAc)_2_ (0.002 mmol) was added to a Schlenk tube. Then HBpin (1.5 equiv.) and THF (2 mL) were added by syringe under an atmosphere of nitrogen. The reaction mixture was stirred at 25 °C for 1 h. After quenching with saturated NH_4_Cl/H_2_O (10 mL), the crude product was extracted with EtOAc (3 × 10 mL). The combined organic phases were dried over anhydrous Na_2_SO_4_ and concentrated under vacuum, and the crude product was purified by column chromatography to afford the desired hydrogenation compound.

### 3.3. General Procedure for Zn-Catalyzed Reduction of Ketones

A mixture of corresponding ketone derivatives (0.2 mmol) and Zn(OAc)_2_ (0.002 mmol) was added to a Schlenk tube. Then HBpin (1.5 equiv.) and THF (2 mL) were added by syringe under an atmosphere of nitrogen. The reaction mixture was stirred at 60 °C for 24 h. After quenching with saturated NH_4_Cl/H_2_O (10 mL), the crude product was extracted with EtOAc (3 × 10 mL). The combined organic phases were dried over anhydrous Na_2_SO_4_ and concentrated under vacuum, and the crude product was purified by column chromatography to afford the desired hydrogenation compound.

### 3.4. General Procedure for Zn-Catalyzed Reduction of Imines

A mixture of corresponding imines (0.2 mmol) and Zn(OAc)_2_ (0.002 mmol) was added to a Schlenk tube. Then HBpin (1.5 equiv.) and THF (2 mL) were added by syringe under an atmosphere of nitrogen. The reaction mixture was stirred at 25 °C for 5 h. After quenching with saturated NH_4_Cl/H_2_O (10 mL), the crude product was extracted with EtOAc (3 × 10 mL). The combined organic phases were dried over anhydrous Na_2_SO_4_ and concentrated under vacuum, and the crude product was purified by column chromatography to afford the desired hydrogenation compound.

## 4. Conclusions

In summary, we successfully demonstrated a method for the synthesis of various alcohols and related derivatives by using low-cost HBpin with commercially available Zn(OAc)_2_ in THF. Good yields and chemoselectivity were obtained for a wide range of substrates, including aldehydes, ketones, *α*,*β*-unsaturated aldehydes/ketones, and imines, showing the great versatility and group tolerance. The current protocol operates under mild conditions and enables the reduction of C=O in good yields and selectivity without the requirement of a hydrolysis reagent (acid or base). Mechanistic studies indicated the formation of active ZnH(OAc) species. Further studies on the mechanism and enantioselective reducing ketones and *α*,*β*-unsaturated ketones using Zn(OAc)_2_/HBpin are undergoing. We believe that the catalytic system is expected to be used for asymmetric hydrogenation of carbonyl compounds.

## Data Availability

Not applicable.

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
