# Peer review of "Zn-Catalyzed Regioselective and Chemoselective Reduction of Aldehydes, Ketones and Imines"

_ijms, 2022, doi:10.3390/ijms232012679_

Round 1

Reviewer 1 Report

This manuscript by Zhang et al. concerns the regio- and chemoselective reduction of aldehydes, ketones and imines by a Zn-catalyst and HBpin. The manuscript is very well written with careful optimization of the reaction conditions and investigation of a wide substrate scope. I will gladly recommend acceptance. Only few minor spelling error were detected and maybe the results could be compared to the previous results a little bit better to give the reader more reference. 

Reviewer 2 Report

In this work Zn-catalyzed regioselective and chemoselective reduction of aldehydes, ketones and imines is described. The work is of interest because presents a rare example of using mild and sustainable HBpin as a reductant for catalytic reduction of carbonyl compounds in the absence of acid or base as hydrolysis reagent. Moreover it was found that the Zn(OAc)2/HBpin combination has been proved to be chemoselective for the C=N reduction of imine analogs. The article looks like a short communication and may be published after minor revision.

Notes:

1. There are some printing mistakes that should be corrected:

            Line 88. The sentence should be started with the capital letter.

            Line 97. “aminomethy” should be changed by “aminomethyl”.

2. By which methods were characterized the obtained products? Were they isolated from reaction mixtures? The yields of products were reported according to isolated products or according to NMR spectra data? Short comment should be added in the text.

3. In the schemes 2, 3 is not necessary to write the amounts (mmol, g) of initial compounds. This information should be presented in Materials and Methods section.

4. Promising application of new obtained results should be added in Conclusions. What areas of science are in demand in this? It should be reflected in Conclusions.
